# Behavioral, health- related and demographic risk factors of death in floods: A case-control study

Arezoo Yari[1], Homa Yousefi Khoshsabegheh[2,3], Yadolah Zarezadeh[1], Ali Ardalan[2], Mohsen Soufi Boubakran[4], Abbas Ostadtaghizadeh[2]*, Mohamad Esmaeil Motlagh[5]*

**1** Social Determinants of Health Research Center, Research Institute for Health Development, Kurdistan University of Medical Sciences, Sanandaj, Iran, **2** Department of Health in Emergencies and Disasters, School of Public Health, Tehran University of Medical Sciences, Tehran, Iran, **3** Disaster Risk Management Office, Ministry of Health and Medical Education, Tehran, Iran, **4** Department of Mechanical Engineering, Urmia University, Urmia, Iran, **5** Department of Pediatrics, Ahvaz Jundishapur, University of Medical Sciences, Ahvaz, Iran

* ostadtaghizadeh@gmail.com (AOT); Dr.motlagh.ms@gmail.com (MEM)

**Data Availability Statement:** Data cannot be shared publicly because of ethics approval limitation. Data are available from the Kurdistan

## Abstract

During the first half of 2019, many provinces of Iran were affected by floods, which claimed the lives of 82 people. The present study aimed to investigate the behavioral, health related and demographic risk factors associated with deaths due to floods. We measured the odds ratio and investigated the contribution and significance of the factors in relation to mortality. This case-control study was conducted in the cities affected by flood in Iran. Data were collected on the flood victims using a questionnaire. Survivors, a member of the flood victim's family, were interviewed. In total, 77 subjects completed the survey in the case group, and 310 subjects completed the survey in the control group. The findings indicated that factors such as the age of less than 18 years, low literacy, being trapped in buildings/cars, and risky behaviors increased the risk of flood deaths. Regarding the behavioral factors, perceived/real swimming skills increased the risk of flood deaths although it may seem paradoxical. This increment is due to increased self confidence in time of flood. On the other hand, skills and abilities such as evacuation, requesting help, and escape decreased the risk of flood deaths. According to the results, the adoption of support strategies, protecting vulnerable groups, and improving the socioeconomic status of flood-prone areas could prevent and reduce the risk of flood deaths.

## Introduction

Floods are a major cause of death among natural disasters [1]. Death is a sever and irreversible consequence of floods [2], and the frequency of deaths due to floods is reported to be noticeably higher compared to other natural disasters [3–5]. According to statistics, flood deaths are on a rising trend throughout the world [6, 7]. The 2018 World Disasters Report indicated that 1,522 floods have occurred in the world within the past decade (2008–2017). Statistics suggest

University of Medical Sciences Ethics Committee (contact via ethiccommittee@muk.ac.ir) for researchers who meet the criteria for access to confidential data.

**Funding:** This study (no: IR.MUK.REC.1398.223.) was conducted with the financial support of the Deputy for Health at Ministry of Health and Medical Education (MOHE) in IRAN. The funding body did not have any other roles in the study design, collection, analysis, and interpretation of data and in writing the manuscript.

**Competing interests:** The authors have declared that no competing interests exist.

that approximately 37% of the two billion people affected by natural disasters in this period were impacted by floods, and 50,312 deaths have been recorded due to floods [8]. The increased rate of floods and the resulting deaths are more significant in Asia, as well as in developing and underdeveloped countries [9]. Iran is one of the world's most flood-prone countries [10]. In the first half of 2019; heavy rains flooded different regions of the country [11], and this unprecedented disaster caused huge damages, such as death, damage and destruction of buildings and vital infrastructures, burden on the healthcare infrastructure, and destruction and damage of properties and assets (e.g., livestock and agricultural lands) [12]. Furthermore, about 10 million people in more than 2,000 cities and towns across the country have been affected by floods (Figs 1 and 2), and more than half a million were evacuated permanently or temporarily from their homes [11]. Among the affected provinces in Iran, 200 towns and 4,304 villages were damaged totally or partially. Furthermore, the complete destruction of 6,000 urban and rural housing units and damage to more than 75,000 urban and rural housing units were recorded as the detriments caused to properties and assets by floods [12].

According to the literature, the causes of flood death could be classified into two general categories, which are the immediate causes of death, such as drowning [13], physical trauma, heart attack, fire, electrocution, over-exertion, and shock, and underlying factors, including hazard-related factors, and individual, environmental, socioeconomic, and managerial factors [9]. The degree of individual vulnerability, health status and behavior of individuals in the areas affected by floods [14–17], and their response to floods [18, 19] are also classified as the underlying factors of flood deaths. In fact, flood deaths result from long-term attitudes, behaviors, decision-making, and community actions [20].

A high percentage of flood deaths in Europe are attributed to high-risk behaviors [19, 21]. In 2002, the World Health Organization (WHO) reported that approximately 40% of the health effects of floods in Europe are associated with high-risk behaviors [22]. In addition, several of these high-risk behaviors during floods are unnecessary [17, 23], such as using vehicles in the United States during floods, which result in flood deaths [21, 24]. In 2002, Egli also stated that in Switzerland, 40% of the 67 flood deaths during 1972–2001 occurred due to the inappropriate behaviors of the victims [25]. A combination of risk factors and individual vulnerabilities has been shown to cause death in floods [13, 19], and the response of people to floods plays a key role in determining the mortality and morbidity of floods [23].

Awareness, behaviors, decision-making, resilience, problem-solving, skills in dealing with floods, and precautionary measures could help reduce flood deaths. Human behavior and decision-making during natural disasters is a complex process. It is the result of a combination of environmental factors and social processes with individual factors and abilities [26]. Prior experience with flood, trust in public protection (government's protective measures), and excessive emotions affect people's behavior during floods. Notably, decisions to respond to threats, the ability to respond, and the evaluation of the costs and effectiveness of methods affect the decisions of individuals on their response to flood risks. Furthermore, beliefs about the effectiveness of preventive behaviors influence precaution measures, and higher risk perception seems to be associated with better self-protection and preventive measures during floods [27].

In addition to perceptual factors, characteristics such as age, gender, marital status, education level, and attitude toward swimming skills may affect people's risk acceptance and risky behaviors during floods [28]. Depending on the features of the region's population, the mortality rate may increase during floods, and the demographic features of the flood-prone area may expose the youth or vulnerable groups (e.g., women, the elderly) to the higher risk of death [23]. Subsequently, protective measures for hazards and adopting protective behaviors should be promoted among the elderly [29, 30]. However, Diakakis et al in 2018 claimed that although

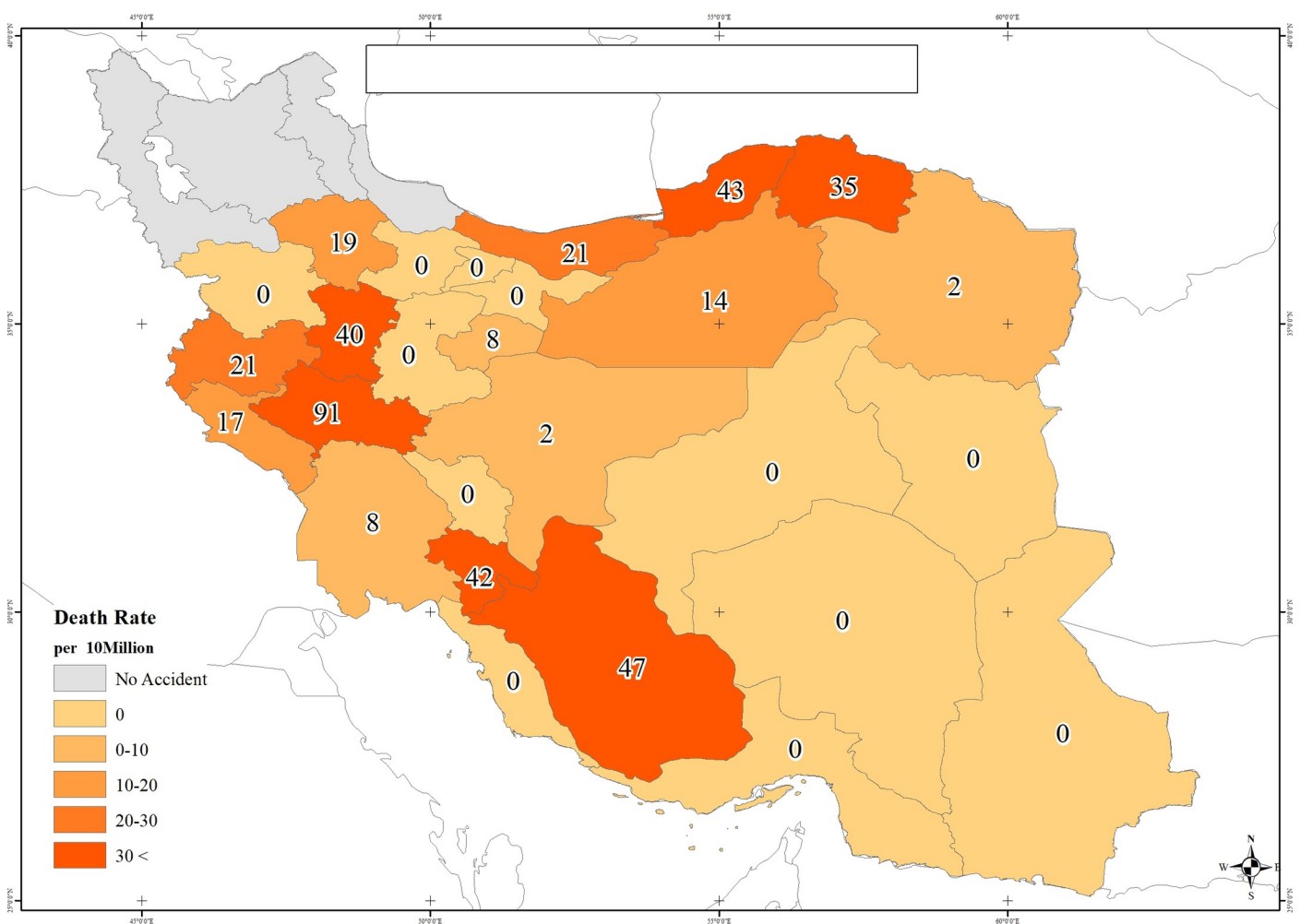

**Fig 1. Flood death rate per 10,000,000 in Iran: 2019.** This map was constructed by the authors using existing data in various sources in Iran's floods in 2019.

the elderly tends to assess their knowledge of floods and flood protection measures more commonly than the youth, age and protective measures have no significant association in this regard [31].

Flood management efficiency is of great importance considering the adverse effects of this natural disaster on the loss of life and economic losses [32]. One of the primary goals of the Sendai framework for disaster risk reduction in 2015–2030 is to reduce deaths in the populations affected by natural hazards. Henceforth, the implementation of risk management policies and appropriate interventions to reduce flood deaths necessitates the assessment of the influential factors in flood deaths [32].

Considering the impact of behavior, health status, and demographic factors on the flood deaths and floods in Iran in 2019 and in line with the main objectives of the Sendai framework, the present study aimed to determine the behavioral, health-related, and demographic risk factors associated with flood deaths in a respective manner. The required data were collected and compared on the individual and behavioral factors of recent flood deaths from the families of the victims and survivors by conducting a survey. Another objective of the research was to

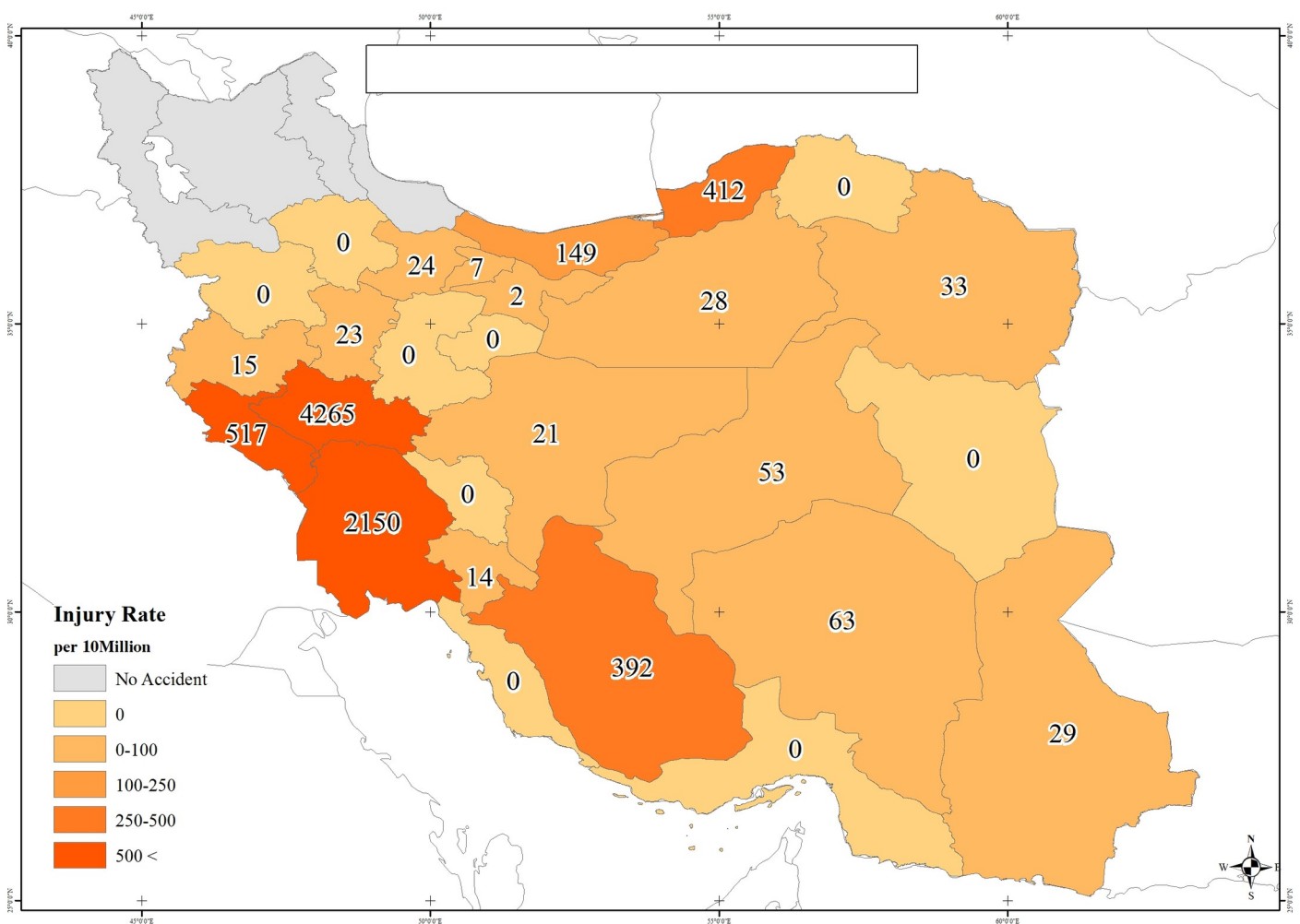

**Fig 2. Flood injury rate per 10,000,000 in Iran: 2019.** This map was constructed by the authors using existing data in various sources in Iran's floods in 2019.

determine the optimal preventive strategies based on the most significant factors in this regard.

## Methods

### Study design and setting

In 2019, several floods occurred in Iran, in which 2,193 people were injured, and 82 died in the affected provinces (Figs 1 and 2). The study areas in the present study were different cities across Iran that had been affected by the deaths occurring in the successive floods in 2019.

In Figs 1 and 2, the provinces that were not affected by the floods are shown in gray, and those affected by the floods are shown in different shades of orange and red. Increased color intensity in the affected provinces (tending to red) indicated the higher rates of deaths and injuries.

In the present study, we considered flood deaths as an outcome to behavioral, health-related, and demographic risk factors, and the investigation was performed with a case-control design. The main objective of the research was to determine the association between these risk factors and the known outcome of floods. A key One of the advantage s of a case-control study

is the simultaneous examination of several risk factors [33]. case-control studies often start with the identification of the case and control groups, followed by the retrospective assessment of the shared risk factors between these groups to determine the main associations [34].

## Sample size and sampling

In the present study, almost all the Iranian records of flood deaths were reviewed, including the records of the Disaster Management Organization, the Governorate, the Ministry of Energy, the Ministry of Health (Public Health and Emergency Operation Center), and Forensic Medicine. In addition, we reviewed the data to ensure the proper confirmation of the recorded death rates, which enables us to attain the demographic data of the flood victims as well, including age, gender, geographical location, home address, flood time, and phone number of the closest family member/other persons. In total, 86 cases of flood deaths were reviewed, which had been recorded in various sources of the floods occurring in Iran during 17th March-29th May 2019.

Based on the mentioned definition of flood death, only 82 deaths could be identified as four cases had died due to other reasons coinciding with the floods. We investigated the behavioral, health-related, and demographic causes of these 82 deaths. During the investigation, the address and phone numbers of two flood victims' relatives were false, and follow-up was not possible. In addition, three relatives of the flood victims were unwilling to partake in the study. Finally, the study was performed using the data of 77 flood victims who had died in Iran's floods in 2019. Approximately 93.9% of the relatives of the flood victims participated in the study.

This retrospective study was conducted with a case-control design and started three months after the floods occurring in Iran after 29th May, 2019. Therefore, the probability of recall bias was low, and the strength of the study was increased by considering the ratio of the controls to the cases to be 4:1. In other words, four controls were selected per each case of death.

## Cases and controls

In the current research, the victims who had died due to direct contact with the floods in 2019 were classified as the case group. If the individual facing the floods had died due to other causes than flood, they would not be considered as cases. The control group included the survivors or living individuals in the flooded areas during the floods that caused the deaths of the flood victims, including the residents of the flooded areas in the recent incidents. To select the controls via population-based sampling, we randomly selected the control samples from the neighbors of the flood victims based on family health dossier number in the Comprehensive Health Services and Health Centers (10 numbers above and below).

## Variable measurements

Flood death was considered the main outcome variable, which pertained to the deaths caused by floods and defined as the deaths in direct association with the floods or a fatality that would not have occurred without a flood. The health authorities provided a list of the deaths in direct association with the floods, with the exposure factors mainly reported to be the demographic/behavioral factors and health status of the individuals, which could increase or decrease the risk of flood death.

The studied demographic variables included age, gender, education level, occupation status, number of the household members, place of residence (by province), place of residence (by city/village), ethnicity, and income and residence status (indigenous/non-indigenous). In this approach, age was classified in to three main categories less than 18 years, 18–58 years, and

more than 58 years. based on the definition of adulthood in the Iranian culture and legal system (i.e., age of 18 years). The age group of 18–58 years was divided in to 10-year intervals, and age of 58 years was considered the age of retirement. Similarly, the education category was determined in line with the educational system of Iran.

The behavioral and health status were examined in association with abilities and skills, general health, alcohol/drug consumption, prior experience, physical conditions, area recognition, timely escape, attention to safety and danger limits, attention to meteorological warnings, situation, and state during the accident. Behavioral and health status variables were categorized based on having or not having a specific abilities and skills or health status. In addition, some of these multi-mode variables were categorized based on the Likert scale or the number of states of that variable.

Data were collected using a reliable and valid questionnaire, which was developed in a dissertation conducted at the Health in Emergencies and Disasters Department of Tehran University of Medical Sciences. The tool is known as the "Questionnaire of the Factors Affecting Flood Deaths in Iran" [35]. The reliability of the scientific indices of the questionnaire items was evaluated psychometrically in the doctorate dissertation. In addition, the empirical validity of the instrument was assessed by factorial analysis using the Kaiser-Meyer- Olkin (KMO) test. In general, the KMO of higher than 0.6–0.7 in the exploratory factor analysis process indicates the suitability of the variables' number and sampling adequacy [36]. In the current research, the KMO coefficient of the questionnaire was higher than 0.6–0.7 in all the constructs, and Bartlett's test was also considered significant. The reliability or internal consistency of the questionnaire was also evaluated based on Cronbach's alpha, which was within the range of 0.75–0.91, demonstrating favorable/excellent internal consistency [36]. In addition, the internal consistency of the questionnaire was confirmed at the Cronbach's alpha coefficient of 0.92. The instrument was used to measure the influential factors in the flood deaths by 33 items and seven constructs; however, we only presented the demographic, behavioral, and health- related factors affecting the flood deaths in Iran's floods in 2019.

The surveyors were trained on helping the participants to complete the questionnaire accurately and providing the necessary explanations if needed. The survey method was the same for the case and control groups. To obtain the case data, the next-of-kin of the flood victims were enrolled in the research, and the participants were preferably a family member of the flood victims with a higher education level and willingness to participate.

## Matching

In the current research, the group-matching method was applied to facilitate the implementation of the survey. Group- matching was also performed for the variable of the areas affected by the floods, and the control samples were selected as a random sample of the affected population. Furthermore, binary logistic regression analysis was used to ensure the thorough control of the confounding variables as it provided the odds ratio (OR) for controlling multiple confounders. The ORs were documented as the adjusted OR, which should be adjusted for other covariates (e.g., confounders) [37].

## Statistical analysis

Descriptive and analytical statistics were used to evaluate and measure the variables by type and category of variables. Moreover, the correlations between the independent variables (i.e., demographic variables, health- related, and behavioral factors) with flood deaths as a dependent variable were evaluated using analytical statistics, such as Chi-square and Fisher's exact test, based on the OR. In these tests, we determined the presence or absence of correlations

between the behavioral, health related factors, and demographic variables with the flood deaths.

In the binary logistic regression analysis, the effects of these factors on the flood deaths were investigated as well. Moreover, binary logistic regression analysis was performed for all the significant variables in Chi-square or Fisher's exact test, and the effects of confounders were also controlled by the binary logistic regression analysis.

## Ethical considerations

The interviews were conducted with the participants' consent. Written informed consent was also obtained from participants after providing the necessary explanations. After explaining the objectives of the study to the participants, they were assured of the confidentiality of the data and observance of ethical principles. The participants were allowed to withdraw from the study at any stage. If they found any of the survey items stressful, they were also allowed to not respond. The time and place of the interviews were determined by the participants so as to make them feel comfortable during the interviews. The study protocol was approved by the Institutional Review Board (IRB) of Kurdistan University of Medical Sciences (ethics code: IR. MUK.REC.1398.223); the IRB follows the stipulated clauses of the Declaration of Helsinki.

## Results

### Descriptive and analytical results

In total, 387 participants were enrolled in the study, including 77 cases and 310 controls. The descriptive and simple analytical results of Chi-square or Fisher's exact test have been presented in the following sections, including demographic variables, behavioral and health-related factors.

### Demographic variables

The mean age of the participants was 38.8±15.14 years, and the male-to-female ratio was 3.83. In total, 23.3% of the participants were university graduates, 26.72% had a high school diploma, and 22.4% were a person's cannot read and write. In terms of the residence, 50.4% of the participants were urban, and 49.6% were rural. In addition, 24.6% of the participants were self-employed, and 18.1% were employees. Notably, 84.5% of the participants lived in the flooded areas. The analytical results regarding the correlations between the behavioral and skill characteristics of the victims of flood death based on Chi-square or Fisher's exact test indicated that age, education level, and occupation status were significantly associated with flood death (P<0.005) (Table 1).

### Behavioral factors

The results of the descriptive analysis regarding the behavioral factors of the participants indicated that 98.7% had not consumed alcohol at the time of the floods. Additionally, 74.1% of the participants had not experienced floods before, and area recognition was 52.1% above average. Attention to safety and danger limits in 51.3% and attention to meteorological warnings in 45.3% of the participants were moderate. However, these rates were higher in the control group compared to the case group.

During the floods, 42.2% of the participants had high self-esteem and 'heroic behaviors', 5.2% had risky behaviors, 3.9% of the participants had walked into or entered the floodwaters (flooded river, bridge or road), and 8.2% had been trapped in a building or a vehicle. Regarding the behavioral factors based on abilities and skills, 32.7% of the participants were able to

**Table 1. Association between demographic factors and flood death in Iran floods in 2019, (n = 387).**

| Demographic Factors | Case(n = 77), n (%) | Control(n = 310), n (%) | Test Result | (P value) OR (95% CI) |
|---|---|---|---|---|
| **Sex** | | | | |
| **Female** | 21(27.3) | 54(17.4) | OR = 0.563 | 0.239–1.079 |
| **Male** | 56(72.7) | 256(82.6) | | |
| **Age Group (Year)** | | | | |
| **<18** | 18(23.4) | 10(3.2) | $X^2$ = 27.52 | 0.000 |
| **18–28** | 11(14.3) | 40(12.9) | | |
| **28–38** | 12(15.6) | 100(32.3) | | |
| **38–48** | 16(20.8) | 82(26.5) | | |
| **48–58** | 12(15.6) | 44(14.2) | | |
| **>58** | 8(10.4) | 34(11.0) | | |
| **Education** | | | | |
| **A person cannot read and write** | 27(35.1) | 50(16.2) | $X^2$ = 17.99 | 0.003 |
| **Primary** | 10(13.0) | 18(5.8) | | |
| **Elementary** | 11(14.3) | 12(3.8) | | |
| **High school** | 6(7.8) | 44(14.2) | | |
| **Diploma** | 10(13.0) | 104(33.5) | | |
| **University** | 13(16.8) | 82(26.5) | | |
| **Occupation** | | | | |
| **Driver** | 4(5.2) | 22(7.1) | Exact = 29.39 | 0.000 |
| **rancher or farmer** | 13(16.9) | 40(12.9) | | |
| **Student (school)** | 14(18.2) | 6(1.9) | | |
| **Student (university)** | 0(0.0) | 6(1.9) | | |
| **Employee** | 10(13.0) | 64(20.6) | | |
| **Self- Employment** | 11(14.3) | 92(29.7) | | |
| **Unemployed** | 5(6.5) | 10(3.2) | | |
| **Housewife** | 14(18.2) | 54(17.4) | | |
| **Retired** | 6(7.8) | 16(5.2) | | |
| **Living Place** | | | | |
| City | 45(58.4) | 144(41.6) | OR = 1.621 | 0.933–2.816 |
| Village | 32(46.5) | 166(53.5) | | |
| **Residency situation** | | | | |
| Resident | 65(84.4) | 262(84.5) | OR = 0.992 | 0.467–2.109 |
| Non-Resident | 12(15.6) | 48(15.5) | | |

evacuate, and 9.5% could swim; on the other hand, 61% of the subjects in the case group had no such skills or abilities. The subjects in the control group had one or more skills, and the lack of skills or abilities was not observed in this group. Furthermore, the ability to request help, escape, and scram was greater in the control group compared to the case group.

The analytical results of Chi-square or Fisher's exact test indicated significant correlations between flood death and timely escape, the observance of safety and danger limits, situation and state, attention to meteorological warnings, and type of abilities and skills (Table 2).

## Health- related factors

The investigation of the health status indicated that 98.3% of the subjects had no specific disabilities or illnesses before the floods and used no special medications. Moreover, the analytical results regarding health status and flood death based on Chi-square or Fisher's exact test indicated no significant correlations between the health-related factors and flood death (Table 3).

**Table 2.** Association between personal and behavioral factors and flood death in Iran floods in 2019, (n = 387).

| Personal and Behavioral Factors | Case(n = 77), n (%) | Control(n = 310), n (%) | Test Result | (P value) OR (95% CI) |
|---|---|---|---|---|
| **Alcohol Consumption** | | | | |
| Yes | 1(1.3) | 4(1.3) | OR = 1.007 | 0.090–11.276 |
| No | 76(98.7) | 306(98.7) | | |
| **Having Flood Experience** | | | | |
| Yes | 17(22.1) | 86(27.7) | OR = 0.738 | 0.388–1.404 |
| No | 60(77.9) | 224(72.3) | | |
| **Proper Escape Ability** | | | | |
| Yes | 16(20.8) | 204(65.8) | OR = 0.505 | 0.265–0.960 |
| No | 61(79.2) | 106(34.2) | | |
| **Knowledge of the flooded area** | | | | |
| Very High | 30(39.0) | 116(37.4) | $X^2$ = 7.40 | 0.116 |
| High | 8(10.4) | 50(16.1) | | |
| Average | 13(16.9) | 82(26.5) | | |
| Low | 8(10.4) | 16(5.2) | | |
| Very Low | 18(23.3) | 46(14.8) | | |
| **Attention to safety principles and settlement risk** | | | | |
| Very High | 16(20.8) | 84(27.1) | $X^2$ = 12.61 | 0.013 |
| High | 8(10.4) | 106(34.1) | | |
| Average | 31(40.3) | 84(27.1) | | |
| Low | 10(13.0) | 20(6.5) | | |
| Very Low | 12(15.6) | 16(5.1) | | |
| **Attention to meteorological warnings** | | | | |
| Very High | 20(26.0) | 68(21.9) | $X^2$ = 15.21 | 0.004 |
| High | 11(14.3) | 80(25.8) | | |
| Average | 18(23.4) | 110(35.5) | | |
| Low | 7(9.1) | 8(2.6) | | |
| Very Low | 21(27.3) | 44(14.2) | | |
| **Individual statues at flood time** | | | | |
| Self-confidence and heroism | 50(64.9) | 96(30.9) | $X^2$ = 29.71 | 0.000 |
| Trapping * | 9(11.7) | 20(6.5) | | |
| Risky behavior * | 8(10.4) | 8(2.6) | | |
| Crossing * | 9(11.7) | 0(0.0) | | |
| Fear and stress | 1(1.3) | 186(60.0) | | |
| **Type of ability and skill** | | | | |
| Swimming | 15(19.5) | 14(4.5) | $X^2$ = 160.58 | 0.000 |
| Evacuation | 3(3.9) | 146(47.1) | | |
| Self- Rescue technique | 2(2.6) | 0(0.0) | | |
| Decision making | 1(1.3) | 48(15.5) | | |
| Help request | 3(3.9) | 58(18.7) | | |
| Escape and get away | 1(1.3) | 44(14.2) | | |
| Being multi-skilled | 5(6.5) | 0(0.0) | | |
| Without skill and ability | 47(61.0) | 0(0.0) | | |

## Binary logistic regression

The results of binary logistic regression analysis indicated that age and literacy level (independent variables) were significantly correlated with flood death. For instance, the age of less than 18 years compared to the age of more than 58 years was observed to increase the risk of death

**Table 3. Association between health-related factors and flood death in Iran floods in 2019, (n = 387).**

| Health-Related Factors | Case(n = 77), n (%) | Control(n = 310), n (%) | Odds Ratio | (95% CI) |
|---|---|---|---|---|
| **Having disease** | | | | |
| Yes | 1(1.3) | 4(1.3) | OR = 2.04 | 0.282–14.765 |
| No | 76(98.7) | 306(98.7) | | |
| **Having Disability** | | | | |
| Yes | 3(3.9) | 2(0.6) | OR = 6.24 | 0.639–61.044 |
| No | 74(96.1) | 308(99.4) | | |
| **Medication Consume** | | | | |
| Yes | 2(2.6) | 4(1.3) | OR = 2.04 | 0.282–14.765 |
| No | 75(97.4) | 306(98.7) | | |

significantly (OR = 9.15; 95% CI: 3.12–11.97; P<0.034). Similarly, illiteracy level (OR = 5.42; 95% CI: 2.13–10.32; P<0.02), low literacy (OR = 3.03; 95% CI: 1.36–13.67; P<0.03), and intermediate literacy (OR = 3.23; 95% CI: 3.26–17.94; P<0.04) also increased the risk of flood death compared to academic education (Table 4).

Considering the significant correlation of the safety and danger limits (independent variables) with the flood deaths, very high attention (OR = 3.15; 95% CI: 2.04–8.30; P<0.016) and high attention (OR = 4.06; 95% CI: 1.03–6.81; P<0.03) reduced the risk of flood deaths compared to inadequate attention to these principles. Individuals' situation and state during the accident (independent variables) also had a significant association with the flood deaths, and trapping in a building or car (OR = 23.61; 95% CI: 6.18–15.36; P<0.017) increased the risk of flood death. In addition, high-risk behaviors such as watching, tacking a picture, curiosity, relief efforts and rescue, retrieving stocks or properties, incaution (OR = 37.92; 95% CI: 12.34–23.37; P<0.02), crossing flooded routes, rivers, roads, and bridges (OR = 19.86; 95% CI: 8.65–19.34; P<02) increased the risk of flood deaths compared to the avoidance of high-risk behaviors due to stress or fear (Table 4).

The type of skills and abilities (independent variables) had a significant correlation with the flood deaths; paradoxically, perceived or real swimming skills (OR = 1.93; 95% CI: 2.16–9.32; P<0.04) increased the risk of flood death compared to the lack of such skills and abilities. On the other hand, evacuation abilities (OR = 14.26; 95% CI: 1.25–8.32; P<0.02), decision-making skills (OR = 26.31; 95% CI: 2.98–15.37; P<0.04), ability to request help (OR = 39.26; 95% CI: 1.34–10.25; P<0.02), and ability to escape (OR = 25.06; 95% CI: 2.63–13.95; P<0.01) decreased the risk of flood death compared to the lack of such skills and abilities. In addition, escape at the time of a flood time (independent variable) had a significant correlation with the flood deaths, while timely escape during the floods (OR = 13.88; 95% CI: 1.23–15.63; P<0.03) reduced the risk of flood death compared lack thereof (Table 4).

## Discussion

This descriptive and analytical study compared the behavioral, health-related factors, and demographic variables between the case and control groups, and the effect and size of the associations were also measured and interpreted using binary logistic regression analysis. The comprehensive survey of the behavioral, health-related factors, and demographic variables clarified whether the recognized risk factors were significantly correlated with flood death. Furthermore, we attempted to assess the significance of the correlation and whether it increased or decreased the risk of flood death.

**Table 4. Beta coefficients and test statistics of the variables used in the logistic regression equation.**

| Variable | B | S. E | df | sig | Exp (B) | (95% CI) for EXP(B) |
|---|---|---|---|---|---|---|
| **Age** | | | | | | |
| <18 | 3.854 | 2.981 | 1 | 0.034 | 9.157 | 3.123–11.974 |
| 18–28 | 1.220 | 1.943 | 1 | 0.530 | 0.295 | 0.007–13.320 |
| 28–38 | 2.249 | 2.497 | 1 | 0.894 | 9.482 | 0.071–12.997 |
| 38–48 | 0.265 | 1.988 | 1 | 0.532 | 1.303 | 0.026–64.188 |
| 48–58 | 1.053 | 1.686 | 1 | 0.129 | 2.867 | 0.105–78.089 |
| **Education** | | | | | | |
| A person cannot read and write | 5.580 | 2.590 | 1 | 0.021 | 5.425 | 2.134–10.326 |
| Primary | 17.313 | 32.24 | 1 | 0.036 | 3.036 | 1.362–13.672 |
| Elementary | 6.026 | 2.309 | 1 | 0.049 | 3.234 | 3.264–17.948 |
| High school | 4.216 | 3.157 | 1 | 0.182 | 5.064 | 0.863–7.177 |
| Diploma | 2.965 | 2.463 | 1 | 0.229 | 6.593 | 0.984–6.440 |
| **Occupation** | | | | | | |
| Driver | 2.697 | 2.419 | 1 | 0.265 | 6.235 | 0.001–7.718 |
| rancher or farmer | 3.383 | 1.784 | 1 | 0.058 | 9.034 | 0.001–1.120 |
| Student (school) | 4.245 | 3.202 | 1 | 0.185 | 5.061 | 0.372–65.354 |
| Student (university) | 12.003- | 25.998 | 1 | 0.356 | 4.014 | 0.000–3.267 |
| Employee | 3.320- | 2.494 | 1 | 0.183 | 21.036 | 0.000–4.798 |
| Self- Employment | 2.798 | 2.098 | 1 | 0.161 | 6.061 | 0.001–1.722 |
| Unemployed | 3.763 | 2.687 | 1 | 0.998 | 0.023 | 0.389–41.798 |
| Retired | 8.315- | 27.88 | 1 | 0.219 | 4.028 | 2.689–69.491 |
| **Attention to safety principles and settlement risk** | | | | | | |
| Very High | -5.432 | 3.120 | 1 | 0.016 | 3.159 | 2.004–8.307 |
| High | -4.768 | 2.226 | 1 | 0.037 | 4.061 | 1.037–6.819 |
| Average | -0.713 | 1.902 | 1 | 0.460 | 4.191 | 0.004–1.307 |
| Low | -1.273 | 2.229 | 1 | 0.519 | 9.014 | 0.000–2.687 |
| **Attention to meteorological warnings** | | | | | | |
| Very High | -1.273 | 2.229 | 1 | 0.568 | 0.280 | 0.004–22.098 |
| High | -4.768 | 2.226 | 1 | 0.032 | 0.008 | 0.000–0.666 |
| Average | -0.713 | 1.902 | 1 | 0.708 | 0.490 | 0.012–20.393 |
| Low | -5.432 | 3.120 | 1 | 0.082 | 0.004 | 0.000–1.982 |
| **Individual statues at flood time** | | | | | | |
| Self-confidence and heroism | 6.297 | 3.351 | 1 | 0.060 | 54.313 | 0.764–38.635 |
| Trapping * | 10.081 | 4.214 | 1 | 0.017 | 23.612 | 6.182–15.369 |
| Risky behavior * | 22.480 | 28.783 | 1 | 0.022 | 37.926 | 12.348–23.377 |
| Crossing * | 21.182 | 92.136 | 1 | 0.024 | 19.865 | 8.651–19.349 |
| **Type of ability and skill** | | | | | | |
| Swimming | 37.502 | 13.042 | 1 | 0.048 | 1.937 | 2.165–9.321 |
| Evacuation | 43.986- | 13.0245 | 1 | 0.021 | 14.267 | 1.256–8.325 |
| Self- Rescue technique | -3.227 | 27.951 | 1 | 0.561 | 7.046 | 0.265–123.569 |
| Decision making | 37.803- | 42.327 | 1 | .048 | 26.315 | 2.984–15.371 |
| Help request | -4.215 | 33.178 | 1 | .027 | 39.261 | 1.348–10.258 |
| Escape and get away | -30.038 | 13.669 | 1 | 0.017 | 25.063 | 2.639–13.951 |
| Being multi-skilled | -45.046 | 15.694 | 1 | 0.997 | 14.621 | 0.436–36.327 |
| **Proper Escape Ability** | | | | | | |
| Yes | -2.631 | 1.233 | 1 | 0.033 | 13.888 | 1.239–15.636 |

Age was observed to be an influential factor in the flood deaths in the present study, the effect of which has also been confirmed the previous studies [4, 38]. The regression analysis in the current research indicated that the age of less than 18 years extremely increased the risk of flood deaths, which is consistent with several studies suggesting that young age is a risk factor for flood deaths [3, 4, 24, 39]. Therefore, the improvement and promotion of the safety of flood-prone areas where the youth reside, such as educational places (e.g., schools and institutions) and recreational/sports complexes could reduce the risk of flood death. In addition, some studies have proposed that aging is a risk factor for flood deaths [20, 3, 40], and it is essential to support and implement appropriate measures to protect vulnerable populations, such as the youth and the elderly [6]. Moreover, families should be advised to plan for the support and protection of vulnerable age groups during floods [41].

Our findings indicated that low education levels were a risk factor for flood deaths, which is consistent with the study by Samir. The mentioned study denoted that increased education level could lead to higher awareness, and the type of employment could also reduce the risk of injuries, as well as flood death [42]. The impact of education on risk reduction is evident [43, 44] and has been emphasized in risk reduction and disaster management policies, particularly for the promotion of the culture of resilience in various communities [45]. Therefore, it could be concluded that higher education levels in flood-prone areas may decrease population vulnerability and flood deaths. Considering the mentioned findings regarding the effects of age and education level on flood deaths, managers and decision-makers should pay special attention to the education and training of the youth based on school learning materials, which may involve the incorporation of survival strategies into the curriculum or better informing of the prevention of flooding through lifestyle changes.

In the present study, attention to safety and danger limits decreased the risk of flood death. Consistently, some studies have shown that disregarding and underrating the risk increases the risk of flood death [5, 19, 46]. Consequently, the principles of safety and danger limits must be implemented in residential, educational, recreational, governmental, and non-governmental places, especially in densely populated areas. Furthermore, informing the public and attracting their attention should be attained by installing signs, banners, and posters, as well as through the media and by other appropriate methods to attract their attention to safety principles and danger limits. This is important because early warning and sending alerts in flood-prone areas may play a key role in the reduction of flood deaths [47].

According to the current research, high-risk behaviors such as staying in cars or buildings when the situation demanded otherwise increased the risk of flood deaths. In this regard, the results of binary logistic regression analysis indicated that trapping in a building or car increased the risk of flood death. Previous studies have also shown that being trapped at home/in a building could increase the risk of flood deaths [48, 23]. However, Brazdova and Riha in 2014 stated that buildings that could protect people from the ruins of floods or the effects of water waves may diminish the risk of flood deaths, which implied the importance of standard buildings to protect people during floods [49]. Therefore, the standardization and strengthening of buildings in flood-prone areas are highly recommended. In these areas, buildings should be designed with proper evacuation operations and escape attempts in the case of trapping in the building during floods.

The results of the regression analysis indicated that high-risk behavior such as watching the flood or tacking pictures, being curious, relief efforts and rescue, and retrieving stocks or properties significantly increased the risk of death. Moreover, the analysis showed that crossing flooded routes, river, roads, and bridges was correlated with the higher risk of flood deaths. Several studies have proposed that relief efforts and rescue [19, 50], entering floodwaters [18] or taking unnecessary risks [49] are among the important influential factors in flood deaths.

Therefore, it could be inferred that high-risk behavior increase the risk of flood death and should be avoided altogether. It also seems that the level of awareness and perceptions toward flood risks are rather low in communities, and the adoption of strategies such as educational interventions, raising public awareness on floods [51, 52], preventive measures, public awareness measures in the prevention phase [53], and increasing preparedness and response are strongly recommended.

According to the current research, skills and abilities affected flood-related mortality. Paradoxically, perceived or real swimming skills increased the risk of flood death. Several studies have also confirmed that the people who try to walk into/swim in floods are more likely to die [18, 19]. Perceived or real skills of swimming may also increase self-esteem and courage in confronting floodwater, thereby increasing the risk of death due to flood because the named skills are insufficient to survive as the nature and behavior of water are dramatically different from pools or lakes. As such, proper training of communities in flood-prone areas regarding the dangers of swimming and walking in floods is paramount. It is also imperative to empower communities in flood-prone areas to improve their decision-making abilities and demand help in the due time during floods. These recommendations are consistent with the policies of disaster risk reduction programs based on education and training to propagate the culture of resilience in various communities [45]. Public education and training for facing floods before and during floods is among the key principles of disaster risk management [19].

In the present study, the results of regression analysis indicated that the possession of abilities and skills such as evacuation, prompt decision-making, requesting help, escape, and scramming could significantly reduce the risk of flood deaths, which implied the high chance of survival depending on the endurance and ability of individuals to find an appropriate shelter [23].

The characteristics of the people affected by floods and their ability to respond and ensure the safety of themselves and their relatives during the flood time could determine their vulnerability to floods [54]. For instance, the people who can make the right decisions, ask for help, act/behave properly [45], and escape in a timely manner [14–16] are at a lower risk of flood death. Considering the notable impact of increasing the self-care knowledge of the residents of flood-prone areas on the reduction of the associated fatalities [55], it is imperative to train people, perform proper exercises, and improve the abilities and skills of the community members in order to reduce the fatalities and damages caused by floods.

Despite the immutability of individual factors, the adoption of support strategies, protection of vulnerable groups, and improvement of socioeconomic factors in flood-prone areas could effectively prevent and decrease the risk of flood deaths. In this regard, we propose several measures, including the adoption of harm reduction policies, improvement and strengthening of socioeconomic support, and promotion of inter-organizational and inter-sectoral cooperation. Furthermore, the optimization of disaster education systems, implementation of educational measures (especially behavioral and skill-based measures), and establishing and strengthening natural hazard management structures with an emphasis on flood and community-based disaster risk reduction are advisable.

## Acknowledgments

This study was carried out with the support of the Social Determinants of Health Research Center, Research Institute for Health Development at Kurdistan University of Medical Sciences and Disaster Risk Management Office, Deputy for Health at Ministry of Health and Medical Education (MOHE) in IRAN. Moreover, the authors would like to thank all

participants of the study who gave us their precious time. We would also like to thank Hassan Elyasi for assisting us in providing the map used in the research project.

## Author Contributions

**Conceptualization:** Yadolah Zarezadeh, Ali Ardalan, Abbas Ostadtaghizadeh, Mohamad Esmaeil Motlagh.

**Data curation:** Homa Yousefi Khoshsabegheh.

**Formal analysis:** Arezoo Yari.

**Funding acquisition:** Homa Yousefi Khoshsabegheh.

**Investigation:** Mohsen Soufi Boubakran.

**Methodology:** Yadolah Zarezadeh, Ali Ardalan, Mohsen Soufi Boubakran.

**Project administration:** Arezoo Yari, Abbas Ostadtaghizadeh, Mohamad Esmaeil Motlagh.

**Resources:** Homa Yousefi Khoshsabegheh, Mohamad Esmaeil Motlagh.

**Software:** Mohsen Soufi Boubakran.

**Supervision:** Abbas Ostadtaghizadeh, Mohamad Esmaeil Motlagh.

**Visualization:** Ali Ardalan.

**Writing – original draft:** Arezoo Yari.

**Writing – review & editing:** Arezoo Yari, Yadolah Zarezadeh, Abbas Ostadtaghizadeh.

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
