## [Decision Letter · Decision Letter 0]

9 Jun 2021

PONE-D-21-06605

How behavior influence flood deaths? A case control study

PLOS ONE

Dear Dr. Abbas,

Thank you for submitting your manuscript to PLOS ONE. After careful consideration, we feel that it has merit but does not fully meet PLOS ONE’s publication criteria as it currently stands. Therefore, we invite you to submit a revised version of the manuscript that addresses the points raised during the review process.

We look forward to receiving your revised manuscript.

Kind regards,

Shah Md Atiqul Haq

Academic Editor

PLOS ONE

Journal Requirements:

5.We note that Figure(s) 1 and 2 in your submission contain map images which may be copyrighted. All PLOS content is published under the Creative Commons Attribution License (CC BY 4.0), which means that the manuscript, images, and Supporting Information files will be freely available online, and any third party is permitted to access, download, copy, distribute, and use these materials in any way, even commercially, with proper attribution. For these reasons, we cannot publish previously copyrighted maps or satellite images created using proprietary data, such as Google software (Google Maps, Street View, and Earth). For more information, see our copyright guidelines: http://journals.plos.org/plosone/s/licenses-and-copyright.

a)   You may seek permission from the original copyright holder of Figure(s) 1 and 2 to publish the content specifically under the CC BY 4.0 license. 

Additional Editor Comments:

Dear authors,

Thank you for submitting this interesting and important article.

I would like to request you to revise the article according to the comments and suggestions of the auditors.

Kind regards,

Reviewers' comments:

Reviewer's Responses to Questions

**Comments to the Author**

1. Is the manuscript technically sound, and do the data support the conclusions?

Reviewer #1: No

Reviewer #2: Yes

2. Has the statistical analysis been performed appropriately and rigorously? 

Reviewer #1: No

Reviewer #2: Yes

3. Have the authors made all data underlying the findings in their manuscript fully available?

Reviewer #1: Yes

Reviewer #2: Yes

4. Is the manuscript presented in an intelligible fashion and written in standard English?

Reviewer #1: No

Reviewer #2: Yes

5. Review Comments to the Author

Reviewer #1: I am not sure whether the Journal guideline follows structured way of abstract (aim, methods, results, conclusion..)

I don’t think the introduction flows as it should. Can they first discuss what Disaster is, specifically flood death the problem at the global level, then continent level, and then Iran. Need more to define problem by giving numbers and trends. And then specifically discuss the regional differences in Iran, and possible reasons for all the problems.

What does mean questionnaire and survey@line 46? Subject is not also appropriate way of explanation. Use standard research language.

You should have justification for this padoxical results either literature driven or data driven justification line 51 to 52.

Sentences from Line 2019-2023 are not clear and I don’t think so it is appropriate way of expression.

The explanations in the text such as surveyor, matching, are not self-explanatory.

The methods of analysis did not elaborated in the appropriate place. There is poor presentation

You should use appropriate place and appropriate description of variables. Avoid the bracket description @line 256.

What was your justification to take only these limited variables?

The paper don't have any theoretical framework as the base for the study

Ethical consideration is considered in the case of health related studies. Please follow the journal guideline for such cases.

The description of explanatory and outcome variables are not consistent.

Category of demographic and other variables should have justification.

Education has six categories with no justification. Illiterate is a person who don’t know. Every person has his/her own indigenous knowledge. You better replace it with cannot read and write.

Avoid the SE,Wald...as you are not used in the description. Use only relevant tests for your analysis Table 4

The majority of your studies are in line with previous findings. What was your contribution/what is new in your study?

There are also repetition of ideas over the paper which needs improvement

English proofreading is needed to provide the expected scientific and right English.

Reviewer #2: The present study aimed to investigate the behavioral and demographic risk factors in the deaths due to flood, starting a case-control study that was conducted in the cities affected by flood in Iran. The author measured the odds ratio and investigated the contribution and significance of the factors in relation to mortality. The required data were collected and compared on the individual and behavioral factors of recent flood deaths from the families of the victims and survivors by conducting a survey.

The introduction presents an excellent discussion on the subject, bringing relevant information and statistics about natural disasters, and, more specifically, flood deaths, and the relationship with the behavior of victims, from various sources, such as World Disasters Report e a World Health Organization (WHO).

Regarding the methodology, the authors randomly selected the control samples from the neighbors of the flood victims based on family health dossier number in the Comprehensive Health Services and Health Centers (10 numbers above and below). About de records, 86 cases of flood deaths were reviewed, which had been recorded in various sources of the floods occurring in Iran during 17th March-29th May 2019, ande the study was

performed using the data of 77 flood victims (and 310 subjects completed the survey in the control group). A set of very interesting variables were selected, on demographic characteristics and behavioral aspects, in addition to statistical validation tests, which contributed to the construction of binary logistic regression. In the models, the effects of these factors on the flood deaths were investigated.

The results start with an excellent descriptive analysis of the demographic characteristics of the victims. The results of binary logistic regression analysis indicated that age and literacy level were significantly correlated with flood death. However, the authors could be more explicit about the relationship that exists between more vulnerable sociodemographic profiles with more precarious housing and vulnerable to flood death, in the context of different types of land use and occupation. I suggest that the author make this discussion, as there is selectivity and socio-spatial inequality, in which residents with less education and resources reside in more vulnerable areas. In fact, the greater propensity of people most vulnerable to flood death is already an indication of this socio-spatial inequality, but it is important to carry out the discussion.

The author commented on the fact that greater education may be related to the promotion of the culture of resilience in various communities, but there may be a relationship between the level of exposure of the dwelling and the sociodemographic profile of the population, even considering the profile of the sample selected. After all, when the authors state that The characteristics of the people affected by floods and their ability to respond and ensure the safety of themselves and their relatives during the flood time could determine their vulnerability to floods, it would be interesting to relativize the place of residence. It is not necessary to incorporate it into the models, but just to contextualize it a little more in the analysis of the results.

Otherwise, the findings indicated that factors such as the age of less than 18 years, low literacy, being trapped in buildings/cars, and risky behaviors increased the risk of flood deaths. According to the results, the adoption of support strategies, protecting vulnerable groups, and improving the socioeconomic status of flood-prone areas could prevent and reduce the risk of flood deaths. These are very interesting results, and represent an important contribution to studies on environmental disasters. I congratulate the authors for the excellent article, with solid theoretical review and robust methodology, and whose opinion I already anticipate is positive, for publication in PLOSONE.

6. PLOS authors have the option to publish the peer review history of their article (what does this mean?). If published, this will include your full peer review and any attached files.

Reviewer #1: **Yes: **Zerihun Yohannes Amare

Reviewer #2: **Yes: **The present study aimed to investigate the behavioral and demographic risk factors in the deaths due to flood, starting a case-control study that was conducted in the cities affected by flood in Iran. The author measured the odds ratio and investigated the contribution and significance of the factors in relation to mortality. The required data were collected and compared on the individual and behavioral factors of recent flood deaths from the families of the victims and survivors by conducting a survey.

The introduction presents an excellent discussion on the subject, bringing relevant information and statistics about natural disasters, and, more specifically, flood deaths, and the relationship with the behavior of victims, from various sources, such as World Disasters Report e a World Health Organization (WHO).

Regarding the methodology, the authors randomly selected the control samples from the neighbors of the flood victims based on family health dossier number in the Comprehensive Health Services and Health Centers (10 numbers above and below). About de records, 86 cases of flood deaths were reviewed, which had been recorded in various sources of the floods occurring in Iran during 17th March-29th May 2019, ande the study was

performed using the data of 77 flood victims (and 310 subjects completed the survey in the control group). A set of very interesting variables were selected, on demographic characteristics and behavioral aspects, in addition to statistical validation tests, which contributed to the construction of binary logistic regression. In the models, the effects of these factors on the flood deaths were investigated.

The results start with an excellent descriptive analysis of the demographic characteristics of the victims. The results of binary logistic regression analysis indicated that age and literacy level were significantly correlated with flood death. However, the authors could be more explicit about the relationship that exists between more vulnerable sociodemographic profiles with more precarious housing and vulnerable to flood death, in the context of different types of land use and occupation. I suggest that the author make this discussion, as there is selectivity and socio-spatial inequality, in which residents with less education and resources reside in more vulnerable areas. In fact, the greater propensity of people most vulnerable to flood death is already an indication of this socio-spatial inequality, but it is important to carry out the discussion.

The author commented on the fact that greater education may be related to the promotion of the culture of resilience in various communities, but there may be a relationship between the level of exposure of the dwelling and the sociodemographic profile of the population, even considering the profile of the sample selected. After all, when the authors state that The characteristics of the people affected by floods and their ability to respond and ensure the safety of themselves and their relatives during the flood time could determine their vulnerability to floods, it would be interesting to relativize the place of residence. It is not necessary to incorporate it into the models, but just to contextualize it a little more in the analysis of the results.

Otherwise, the findings indicated that factors such as the age of less than 18 years, low literacy, being trapped in buildings/cars, and risky behaviors increased the risk of flood deaths. According to the results, the adoption of support strategies, protecting vulnerable groups, and improving the socioeconomic status of flood-prone areas could prevent and reduce the risk of flood deaths. These are very interesting results, and represent an important contribution to studies on environmental disasters. I congratulate the authors for the excellent article, with solid theoretical review and robust methodology, and whose opinion I already anticipate is positive, for publication in PLOSONE.

---

## [Author Response · Author response to Decision Letter 0]

22 Oct 2021

PLOS ONE JOURNAL 

Dear Editor 

Thank you very much for considering our manuscript for publication. We appreciate the comments made by the respected Assistant Editor. 

We carefully read the comments and considered the comments and made changes and /or made our manuscript clearer accordingly. 

With respect to the opinions of Assistant Editor, the article was revised. Please note that all changes are done in the main text. 

Third Editor Comment:

1. Thank you for letting us know that your data is confidential and cannot be publicly shared. According to PLOS policy, data that cannot be publicly shared should be made available on request through an ethics or data access committee to researchers who meet the criteria for access to confidential data.

Please update your data availability statement with the name and contact details of the local ethics committee or local data protection manager that will be responsible for data access. For more information on sharing sensitive data, please see our website: https://journals.plos.org/plosone/s/data-availability#loc-human-research-participant-data-and-other-sensitive-data

Response: Thanks for your comment. Based on your comment we update the data availability statement. Also, we added the data availability statement section with the name and contact details of the local ethics committee upon our manuscript. the data availability statement section is shown in red color after acknowledgment, page 23, line 467-471. 

Second Editor Comments:

1.Can you please upload an additional copy of your revised manuscript that does not contain any tracked changes or highlighting as your main article file. This will be used in the production process if your manuscript is accepted. Please amend the file type for the file showing your changes to Revised Manuscript w/tracked changes. Please follow this link for more information: http://blogs.PLOS.org/everyone/2011/05/10/how-to-submit-your-revised-manuscript/

Response: Thanks for your comment. Based on your comment we uploaded an additional copy of revised manuscript that does not contain any tracked changes or highlighting as main article file. 

2. Thank you for updating your data availability statement to indicate that the data will be made available on request. However, please note that it is not acceptable for the only contact for data access to be an author of the paper. The data should be made available through an institutional data access or ethics committee.

Response: Thanks for your comment. Based on your comment we revised data availability statement in cover letter. Edited sentences are highlighted in revised cover letter. 

First Editor Comments:

Response: Thanks for your comment. Based on your comment we revised the style of the article, also we revised the title authors affiliations. Revised title page are abvious in track change file. 

Response: Thanks for your comment. Based on your comment we revised Ethical Considerations in Method section. Edited sentences are highlighted in red color in Ethical Considerations section in Method. 

Response: Thanks for your comment. Based on your comment we revised cover letter. Edited sentences are highlighted in revised cover letter. 

Response: Thanks for your comment. Based on your comment we added ORCID-ID for the corresponding author to his Editorial Manager account. 

5.We note that Figure(s) 1 and 2 in your submission contain map images which may be copyrighted. All PLOS content is published under the Creative Commons Attribution License (CC BY 4.0), which means that the manuscript, images, and Supporting Information files will be freely available online, and any third party is permitted to access, download, copy, distribute, and use these materials in any way, even commercially, with proper attribution. For these reasons, we cannot publish previously copyrighted maps or satellite images created using proprietary data, such as Google software (Google Maps, Street View, and Earth). For more information, see our copyright guidelines: http://journals.plos.org/plosone/s/licenses-and-copyright

a) You may seek permission from the original copyright holder of Figure(s) 1 and 2 to publish the content specifically under the CC BY 4.0 license. 

Response: Thanks for your comment. The maps were created based on the information used in the article and by the research team and were not copied from anywhere. This issue is also mentioned below the maps.

Review Comments to the Author:

Reviewer #1: I am not sure whether the Journal guideline follows structured way of abstract (aim, methods, results, conclusion.)

Response: Thanks for your comment. Based on your comment we changed the abstract structure. We removed unnecessary heading including: background, methods, results, conclusions. these changed are obvious in the abstract section. 

2- I don’t think the introduction flows as it should. Can they first discuss what Disaster is, specifically flood death the problem at the global level, then continent level, and then Iran. Need more to define problem by giving numbers and trends. And then specifically discuss the regional differences in Iran, and possible reasons for all the problems.

Response: Thanks for your comment. based on your comment, we changed the introduction flows: flood disaster and death, flood problem at global level, the continent level and then Iran. Also, we added the numbers and related trends to this section. These changes and new sentences are shown in red color in the introduction section. 

3- What does mean questionnaire and survey@line 46? Subject is not also appropriate way of explanation. Use standard research language.

Response: Thanks for your comment. based on your comment, based on your comment we added new explanation to this section. These changes and new sentences are shown in red color in the abstract section, page 2, line 39.

4- You should have justification for this padoxical results either literature driven or data driven justification line 51 to 52.

Response: Thanks for your comment. In order to justify the paradoxical results, we added new explanations to this section. These changes and new sentences are shown in red color in the abstract section, page 2, line 45. 

5- Sentences from Line 2019-2023 are not clear and I don’t think so it is appropriate way of expression.

Response: Thanks for your comment. based on your comment we revised this section. These new explanations are shown in red color in the variable measurement section of method, page …., line …. 

6- The explanations in the text such as surveyor, matching, are not self-explanatory.

Response: Thanks for your comment. Based on your comment and for self-explanatory we revised this section. These new explanations are shown in red color in the variable measurement and matching sections of method, page 8-10.

7- The methods of analysis did not elaborated in the appropriate place. There is poor presentation

You should use appropriate place and appropriate description of variables. Avoid the bracket description @line 256.

Response: Thanks for your comment. based on your comment. We replaced some sections of the method and the order of the section is now as follows: Study Design and Setting, Sample Size and Sampling, Cases and Controls, Variable Measurements, Matching, Statistical Analysis, Ethical Considerations. Also, we remove the bracket description and revised these descriptions. These changes are obvious in tracked change file and shown in red color in revised file of the article. 

8- What was your justification to take only these limited variables?

Response: Thanks for your comment. Based on the purpose of this study, all demographic and behavioral variables and health status factors affecting death have been studied in this study. In fact, we have determined this relationship based on a valid and reliable tool.

9- The paper don't have any theoretical framework as the base for the study.

Response: Thanks for your comment. Based on your comment we added new explanation related to theoretical framework as the base for the study in the Study Design and Setting of method section, page 6. Line 143-150. 

10- Ethical consideration is considered in the case of health-related studies. Please follow the journal guideline for such cases.

Response: Thanks for your comment. Based on journal guideline we revised Ethical Consideration for our study. the new sentences are shown by red color in the Ethical Consideration in page 11. 

11- The description of explanatory and outcome variables are not consistent.

Response: Thanks for your comment. Based on your comment, we changed the description of explanatory and outcome variables to make it more transparent. Also, we added new explanation to this section. These changes and new explanations are shown in red color in the variable measurement section in method, page 8-9.

12- Category of demographic and other variables should have justification.

Response: Thanks for your comment. Based on your comment we added new explanation related to justification of demographic and other variables in binary regression model. These new explanations are shown in red color in the variable measurement section in method, page 8-9.

13- Education has six categories with no justification. Illiterate is a person who don’t know. Every person has his/her own indigenous knowledge. You better replace it with cannot read and write.

Response: Thanks for your comment. Based on your comment we replaced Illiterate with a person cannot read and write. These replacements are shown in red color in different places including: table 1, table 4 and demographic variable in result section. 

14- Avoid the SE,Wald...as you are not used in the description. Use only relevant tests for your analysis Table 4

Response: Thanks for your comment. Based on your comment we removed the SE,Wald section from table 4. These changes are obvious in table 4 in result section. 

15- The majority of your studies are in line with previous findings. What was your contribution/what is new in your study?

Response: Thanks for your comment. This was an original study conducted using a rigorous methodology with novel findings. In this case-control study, we were able to quantitatively measure the risk of independent variables (odds ratio) in flood death. The findings of this study are new and practical that cannot be found in other studies. Details and tables are provided in the Findings section.

16- There are also repetition of ideas over the paper which needs improvement

Response: Thanks for your comment. Based on your comment we delete the repetitions of ideas over the paper which needs improvement, these changes and deleted sentences are specified in the crack change file. 

17- English proofreading is needed to provide the expected scientific and right English.

Response: Thanks for your comment. Based on your comment we proofreading the article by native editor, these changes and deleted sentences are specified in the crack change file. Also, we will send the native edit certificate with other documents.

Reviewer #2: The present study aimed to investigate the behavioral and demographic risk factors in the deaths due to flood, starting a case-control study that was conducted in the cities affected by flood in Iran. The author measured the odds ratio and investigated the contribution and significance of the factors in relation to mortality. The required data were collected and compared on the individual and behavioral factors of recent flood deaths from the families of the victims and survivors by conducting a survey.

The introduction presents an excellent discussion on the subject, bringing relevant information and statistics about natural disasters, and, more specifically, flood deaths, and the relationship with the behavior of victims, from various sources, such as World Disasters Report e a World Health Organization (WHO).

Regarding the methodology, the authors randomly selected the control samples from the neighbors of the flood victims based on family health dossier number in the Comprehensive Health Services and Health Centers (10 numbers above and below). About de records, 86 cases of flood deaths were reviewed, which had been recorded in various sources of the floods occurring in Iran during 17th March-29th May 2019, ande the study was

performed using the data of 77 flood victims (and 310 subjects completed the survey in the control group). A set of very interesting variables were selected, on demographic characteristics and behavioral aspects, in addition to statistical validation tests, which contributed to the construction of binary logistic regression. In the models, the effects of these factors on the flood deaths were investigated.

The results start with an excellent descriptive analysis of the demographic characteristics of the victims. The results of binary logistic regression analysis indicated that age and literacy level were significantly correlated with flood death. However, the authors could be more explicit about the relationship that exists between more vulnerable sociodemographic profiles with more precarious housing and vulnerable to flood death, in the context of different types of land use and occupation. I suggest that the author make this discussion, as there is selectivity and socio-spatial inequality, in which residents with less education and resources reside in more vulnerable areas. In fact, the greater propensity of people most vulnerable to flood death is already an indication of this socio-spatial inequality, but it is important to carry out the discussion.

The author commented on the fact that greater education may be related to the promotion of the culture of resilience in various communities, but there may be a relationship between the level of exposure of the dwelling and the sociodemographic profile of the population, even considering the profile of the sample selected. After all, when the authors state that The characteristics of the people affected by floods and their ability to respond and ensure the safety of themselves and their relatives during the flood time could determine their vulnerability to floods, it would be interesting to relativize the place of residence. It is not necessary to incorporate it into the models, but just to contextualize it a little more in the analysis of the results.

Otherwise, the findings indicated that factors such as the age of less than 18 years, low literacy, being trapped in buildings/cars, and risky behaviors increased the risk of flood deaths. According to the results, the adoption of support strategies, protecting vulnerable groups, and improving the socioeconomic status of flood-prone areas could prevent and reduce the risk of flood deaths. These are very interesting results, and represent an important contribution to studies on environmental disasters. I congratulate the authors for the excellent article, with solid theoretical review and robust methodology, and whose opinion I already anticipate is positive, for publication in PLOSONE.

6. PLOS authors have the option to publish the peer review history of their article (what does this mean?). If published, this will include your full peer review and any attached files.

Do you want your identity to be public for this peer review? For information about this choice, including consent withdrawal, please see our Privacy Policy.

Reviewer #1: Yes: Zerihun Yohannes Amare

Reviewer #2: Yes: The present study aimed to investigate the behavioral and demographic risk factors in the deaths due to flood, starting a case-control study that was conducted in the cities affected by flood in Iran. The author measured the odds ratio and investigated the contribution and significance of the factors in relation to mortality. The required data were collected and compared on the individual and behavioral factors of recent flood deaths from the families of the victims and survivors by conducting a survey.

The introduction presents an excellent discussion on the subject, bringing relevant information and statistics about natural disasters, and, more specifically, flood deaths, and the relationship with the behavior of victims, from various sources, such as World Disasters Report e a World Health Organization (WHO).

Regarding the methodology, the authors randomly selected the control samples from the neighbors of the flood victims based on family health dossier number in the Comprehensive Health Services and Health Centers (10 numbers above and below). About de records, 86 cases of flood deaths were reviewed, which had been recorded in various sources of the floods occurring in Iran during 17th March-29th May 2019, ande the study was

performed using the data of 77 flood victims (and 310 subjects completed the survey in the control group). A set of very interesting variables were selected, on demographic characteristics and behavioral aspects, in addition to statistical validation tests, which contributed to the construction of binary logistic regression. In the models, the effects of these factors on the flood deaths were investigated.

The results start with an excellent descriptive analysis of the demographic characteristics of the victims. The results of binary logistic regression analysis indicated that age and literacy level were significantly correlated with flood death. However, the authors could be more explicit about the relationship that exists between more vulnerable sociodemographic profiles with more precarious housing and vulnerable to flood death, in the context of different types of land use and occupation. I suggest that the author make this discussion, as there is selectivity and socio-spatial inequality, in which residents with less education and resources reside in more vulnerable areas. In fact, the greater propensity of people most vulnerable to flood death is already an indication of this socio-spatial inequality, but it is important to carry out the discussion.

The author commented on the fact that greater education may be related to the promotion of the culture of resilience in various communities, but there may be a relationship between the level of exposure of the dwelling and the sociodemographic profile of the population, even considering the profile of the sample selected. After all, when the authors state that The characteristics of the people affected by floods and their ability to respond and ensure the safety of themselves and their relatives during the flood time could determine their vulnerability to floods, it would be interesting to relativize the place of residence. It is not necessary to incorporate it into the models, but just to contextualize it a little more in the analysis of the results.

Otherwise, the findings indicated that factors such as the age of less than 18 years, low literacy, being trapped in buildings/cars, and risky behaviors increased the risk of flood deaths. According to the results, the adoption of support strategies, protecting vulnerable groups, and improving the socioeconomic status of flood-prone areas could prevent and reduce the risk of flood deaths. These are very interesting results, and represent an important contribution to studies on environmental disasters. I congratulate the authors for the excellent article, with solid theoretical review and robust methodology, and whose opinion I already anticipate is positive, for publication in PLOSONE.

---

## [Decision Letter · Decision Letter 1]

16 Dec 2021

Behavioral, Health- Related and Demographic Risk Factors of Death in Floods: A Case-Control Study

PONE-D-21-06605R1

Dear Dr. Abbas Ostadtaghizadeh,

We’re pleased to inform you that your manuscript has been judged scientifically suitable for publication and will be formally accepted for publication once it meets all outstanding technical requirements.

Kind regards,

Shah Md Atiqul Haq

Academic Editor

PLOS ONE

Additional Editor Comments (optional):

Dear authors,

Your article is now accepted.

Reviewers' comments:

Reviewer's Responses to Questions

**Comments to the Author**

1. If the authors have adequately addressed your comments raised in a previous round of review and you feel that this manuscript is now acceptable for publication, you may indicate that here to bypass the “Comments to the Author” section, enter your conflict of interest statement in the “Confidential to Editor” section, and submit your "Accept" recommendation.

Reviewer #2: All comments have been addressed

2. Is the manuscript technically sound, and do the data support the conclusions?

Reviewer #2: Yes

3. Has the statistical analysis been performed appropriately and rigorously? 

Reviewer #2: Yes

4. Have the authors made all data underlying the findings in their manuscript fully available?

Reviewer #2: Yes

5. Is the manuscript presented in an intelligible fashion and written in standard English?

Reviewer #2: Yes

6. Review Comments to the Author

Reviewer #2: The authors complied with all the comments made by the reviewers. Therefore, my recommendation is in favor of publication.

7. PLOS authors have the option to publish the peer review history of their article (what does this mean?). If published, this will include your full peer review and any attached files.

Reviewer #2: No

---

## [Editor Report · Acceptance letter]

23 Dec 2021

PONE-D-21-06605R1 

Behavioral, Health- Related and Demographic Risk Factors of Death in Floods: A Case-Control Study 

Dear Dr. Ostadtaghizadeh:

I'm pleased to inform you that your manuscript has been deemed suitable for publication in PLOS ONE. Congratulations! Your manuscript is now with our production department. 

Kind regards, 

on behalf of

Dr. Shah Md Atiqul Haq 

Section Editor

PLOS ONE